# Changes in searching behaviour of CSL transcription complexes in Notch active conditions

Sarah Baloul*, Charalambos Roussos*, Maria Gomez-Lamarca, Leila Muresan, Sarah Bray

**During development cells receive a variety of signals, which are of crucial importance to their fate determination. One such source of signal is the Notch signalling pathway, where Notch activity regulates expression of target genes through the core transcription factor CSL. To understand changes in transcription factor behaviour that lead to transcriptional changes in Notch active cells, we have probed CSL behaviours in real time, using in vivo Single Molecule Localisation Microscopy. Trajectory analysis reveals that Notch-On conditions increase the fraction of bound CSL molecules, but also the proportion of molecules with exploratory behaviours. These properties are shared by the co-activator Mastermind. Furthermore, both CSL and Mastermind, exhibit characteristics of local exploration near a Notch target locus. A similar behaviour is observed for CSL molecules diffusing in the vicinity of other bound CSL clusters. We suggest therefore that CSL acquires an exploratory behaviour when part of the activation complex, favouring local searching and retention close to its target enhancers. This change explains how CSL can efficiently increase its occupancy at target sites in Notch-On conditions.**

## Introduction

Animal development is shaped by the activity of several highly conserved signalling pathways (Ingham & McMahon, 2001; Wiese et al, 2018; Herrera & Bach, 2019). One such pathway is the Notch signalling pathway which, through the core transcription factor CSL (CBF1, Suppressor of Hairless, Lag), activates cohorts of target genes to regulate decisions during the development of many tissues (Artavanis-Tsakonas et al, 1999; Bray, 2006, 2016; Kopan & Ilagan, 2009; Kovall & Blacklow, 2010). As the dysfunction of the pathway has been linked with various human pathological contexts (Ntziachristos et al, 2014; Nowell & Radtke, 2017), there are stakes in understanding the molecular mechanisms that enable CSL complexes to find and interact with their target loci to implement Notch pathway activity.

The primary link between cell signalling and gene regulation are transcription factors, the behaviour of which is affected by signal activation through many different mechanisms. Several of these involve regulated nuclear translocation, enabled by a release from cytoplasmic tethers or by association with a co-factor (Weidemüller et al, 2021). In the case of the Notch pathway, activation brings about cleavage of the Notch receptor to release the intracellular domain, NICD, which translocates into the nucleus (Schroeter et al, 1998; Gordon et al, 2007). There it forms an activator complex with CSL and co-activator Mastermind (Petcherski & Kimble, 2000; Nam et al, 2006; Wilson & Kovall, 2006). However, in the absence of Notch activity, CSL is present in repressor complexes, partnered with a variety of co-repressors that include co-repressor Hairless in *Drosophila* (Morel et al, 2001; Barolo et al, 2002; Yuan et al, 2016). The activator and repressor complexes co-exist in the nucleus and, although both types of complex have the same DNA binding properties in vitro (Wilson & Kovall, 2006; Bianco et al, 2010), signalling leads to enrichment of CSL activation complexes at target enhancers (Krejčí & Bray, 2007; Castel et al, 2013; Wang et al, 2014; Gomez-Lamarca et al, 2018; DeHaro-Arbona et al, 2023). One question therefore is how the properties of CSL are altered to favour recruitment in Notch active conditions and what mechanisms influence the target search and binding processes.

Knowledge about the fundamental diffusive motion of transcription factors has been proven informative for understanding their recruitment to and interactions with target enhancers in many different pathways and systems (Izeddin et al, 2014; Woringer & Darzacq, 2018; Stavreva et al, 2019; Hansen et al, 2020; Tang et al, 2022; Mazzocca et al, 2023). Live-cell imaging has revealed that interactions with cognate binding sites typically last seconds and that transcription factors exhibit a range of dynamic behaviours. For example, both CTCF and CBX2 manifest properties consistent with local confined motion that could reduce the time taken to find their target regulatory elements (Hansen et al, 2020; Kent et al, 2020). In addition, the transcription factors FOXA1 and SOX2 exhibited distinct chromatin-scanning dynamics that related to their searching behaviours (Lerner et al, 2023). It is therefore possible that CSL activation complexes acquire different searching

---

Physiology Development and Neuroscience, Cambridge Advanced Imaging Centre, University of Cambridge, Cambridge, UK

Correspondence: lam94@cam.ac.uk; sjb32@cam.ac.uk
Maria Gomez-Lamarca's present address is Departamento de Biología Celular, Instituto de Biomedicina de Sevilla (IBiS), Hospital Universitario Virgen del Rocío/CSIC/ Universidad de Sevilla, Seville, Spain
*Sarah Baloul and Charalambos Roussos are joint first authors

 

or scanning behaviours which enable them to become efficiently recruited to target enhancers.

To investigate the diffusive properties of CSL in real-time and measure the changes caused by Notch activation we performed Single Molecule Localisation Microscopy, SMLM (Mazza et al, 2012; Gebhardt et al, 2013), of endogenous CSL, co-activator Mastermind, and co-repressor Hairless in live tissues. Our results show that Notch activation leads to increased exploration and more anisotropic CSL behaviour, consistent with the properties we detect for Mastermind, its partner in the activation complex. The increase in CSL exploratory behaviour is more apparent in proximity to a target locus, *E(spl)-C*, where it becomes locally restricted. Moreover, as the distribution of CSL dynamics is not spatially homogeneous - with bound molecules clustered in different parts of the nucleus and anisotropic behaviour being more common near these clusters—we suggest that in Notch-On conditions, the local exploration that CSL activation complexes undergo near *E(spl)-C* occurs at many different regions in the nucleus, and is a mechanism that enables them to locate their target sites more efficiently.

## Results

### Diffusion dynamics of CSL in Notch-Off conditions

We first set out to characterize the properties and analyse the kinetic dynamics of CSL in nuclei without endogenous Notch pathway activity using SMLM. We generated a Halo-tagged CSL (Halo::CSL), expressed at endogenous levels from a genomic rescue construct (Gomez-Lamarca et al, 2018) and tracked its behaviour in *Drosophila* larval salivary glands, whose large nuclei and polytene chromosome make them amenable for live imaging of transcription factors (Lis, 2007). To resolve single molecules of CSL, salivary glands were incubated with a limiting concentration of the Halo ligand TMR and imaged with 10 and 50 ms exposure times for 3–6 min (Video 1). For comparison, we also carried out SMLM of Halo-labelled Histone H2AV in the same tissue (Video 2).

Following SMLM movie acquisition, we performed single molecule localisation using a Gaussian fitting-based approach (Ovesný et al, 2014). Consecutive frame localisations likely to represent the same molecule were then linked into trajectories, by employing a Multiple Hypotheses Tracking Icy-plugin (Chenouard et al, 2013) (Fig 1A). As the molecules are likely to transition through a range of behaviours reflecting their interactions with the chromatin, the trajectories were then analysed to distinguish their properties using two different methods, which ultimately yielded similar conclusions. The first approach deployed a variational Bayesian treatment of Hidden Markov models (vbSPT) (Persson et al, 2013), where the diffusion constants of a particle trajectory can switch randomly according to a Markov process. The analysis uses a specified number of diffusive states and having set a ceiling, the number of coefficients/populations is chosen via model selection. The second approach, DDMap, assumes that trajectories (or subtrajectories) undergo diffusion, subdiffusion or super-diffusion movements (Saxton, 1993; Briane et al, 2018; Salomon et al, 2020). The software assigns a different diffusion coefficient to each

trajectory and the value of the diffusion coefficients is not constrained in any way. For more information on the trajectory analysis see the Materials and Methods section.

As a starting point, we compared the behaviour of CSL in Notch-Off conditions to the histone H2AV. Because the majority of histones are integrated into nucleosomes and expected to display bound behaviour we first limited the population number for vbSPT analysis to two, which yielded a slow and a fast state. With 10 ms exposure times, the slow-moving population accounted for 34% of CSL trajectories in comparison to 62% of H2AV trajectories, indicating that a relatively smaller proportion of CSL are retained on the chromatin. With the 50 ms exposure time, a proportion of the fastest moving molecules would not be detected as they become blurred. As expected, the average diffusion coefficient of the fastest molecules population was concomitantly reduced from 1.192 to 0.367 $\mu m^2$/s for CSL (Fig 1B and Video 3), indicating that the longer exposure times will give more scope for teasing apart different properties among the molecules whose motions are slowed by their interactions with the chromatin.

In order to probe into possible exploratory behaviours manifest by CSL in Notch-Off conditions, a more fine-grained vbSPT analysis of trajectories from the 50 ms exposure times was carried out using four diffusion states, D1–D4, from slowest to fastest diffusion constant. The four states were selected considering the likelihood estimation from the model selection in vbSPT, which indicated that fewer states were less likely. Trajectories were assigned in these four populations, according to their percentage time in each state and their position then mapped onto the nuclei (Fig 1C and D). Some regions were more strongly enriched for D4 fast-moving molecules and likely correspond to regions with lower density of chromatin (Fig 1C). Others were biased towards slower moving molecules although these were widely distributed in clusters throughout. To further characterize the four populations, we analysed displacement (Fig 1F) and angle distributions (Fig 1G). Displacement distributions inform about the range of motion, with small displacements indicating a restricted behaviour, as was the case for the slowest populations, D1 and D2 (Fig 1F). The distribution of angles indicates the extent to which the movement differs from the uniform distribution expected for regular Brownian motion. For example, an accumulation of angles around 180°, known as backwards anisotropy, occurs when molecules are more likely to move backwards than forwards. This compact behaviour suggests constraint or "trapping" of the molecules and was more enriched in the D1 and D2 populations (Ben-Avraham & Havlin, 2000; Liao et al, 2012; Burov et al, 2013; Izeddin et al, 2014).

By combining all these parameters together (diffusion coefficients, displacement, and angle analyses) we identified behaviours characteristic of each population. Those with the smallest diffusion coefficients, D1 and D2, have properties of bound/trapped molecules, with a high degree of anisotropy and little spatial displacement (Fig 1D–G). They accounted for 6% and 19% of the total CSL molecules respectively (Fig 1H). The slightly longer displacements exhibited by D2 molecules (Fig 1F) likely reflect an association with more mobile chromatin and/or local exchange on and off the DNA. The fastest population, D4, accounting for 48% of trajectories, showed properties of more freely diffusing molecules, with less backwards angular anisotropy and large spatial displacements (Fig 1D–G). As noted

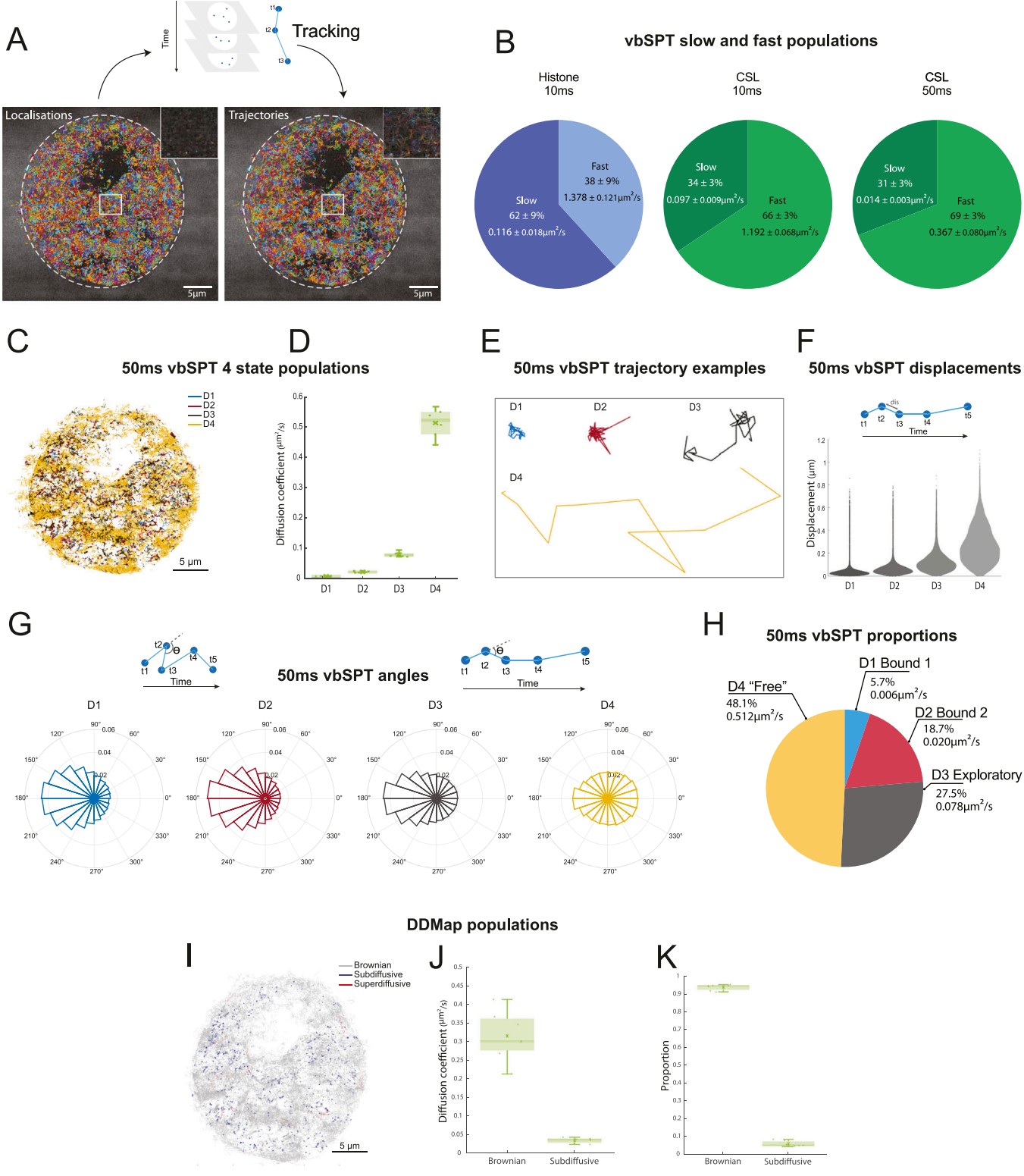

**Figure 1. Characterising different types of CSL behaviour in Notch-Off conditions using Single Molecule Localisation Microscopy analysis pipeline.**
**(A)** Example of single molecule localisations of CSL::Halo acquired at 50 ms exposure (left) and of the trajectories defined by tracking the same molecule in consecutive time frames (right), as illustrated in the schematic above. Dotted lines outline nucleus and insets are 10x zoom of the representative region annotated by the white rectangle. **(B)** Mean proportion of Histone (H2av) and CSL trajectories belonging to each of two populations (slow and fast) defined by vbSPT analysis from 10 and 50 ms exposure times with mean diffusion coefficients. 10 ms histone n = 3 nuclei, 22,820 trajectories; 10 ms CSL, n = 3 nuclei, 35,992 trajectories; 50 ms CSL n = 7 nuclei, 72,980 trajectories. **(C)** Nuclear localisations of single molecule CSL trajectories (representative 50 ms experiment), coloured according to vbSPT diffusion state: blue D1, red D2,

above, many freely/fast moving molecules in the nucleus will not have been captured with the 50 ms regime. Nevertheless, we refer to the D4 population as "free" as their movements are not demonstrably constrained (Fig 1D–G). Finally, molecules in the D3 population have intermediate characteristics, with less backwards anisotropy and more displacement than the bound populations (Fig 1D–G). These account for 27% and their characteristics suggest an exploratory behaviour, with a mixture of binding/trapping and freer movements (Fig 1E and H).

For comparison we utilised DDMap analysis which assigned a different diffusion coefficient to each trajectory (Fig 1J) and, using a nonparametric three-decision statistical test, classified them as Brownian, sub-diffusive or super-diffusive (Fig 1I). Only 6% of CSL trajectories were classified as sub-diffusive, a proportion close to those classified as "bound" D1 by the vbSPT analysis. The majority of CSL molecules (94%) were identified as Brownian, suggesting that all other CSL molecules have some element of Brownian motion associated with their trajectories (Fig 1K). Not unexpectedly, a negligible number of trajectories were classified as super-diffusive.

Together the results demonstrate that a relatively small proportion of CSL is stably associated with chromatin in the absence of Notch activity. The majority of CSL molecules are more transiently associated with chromatin and/or freely moving.

### Notch activity leads to increased binding and exploratory behaviour of CSL

To investigate the effect of Notch activity on CSL we acquired SMLM movies at 50 ms exposure in Notch-On conditions (Video 4), achieved by expressing a constitutively active form of Notch in salivary gland cells. This gives rise to tissues with a steady state "Notch-On," in which known target genes are transcriptionally active (Gomez-Lamarca et al, 2018; DeHaro-Arbona et al, 2023), that can be compared with the Notch-Off tissues where those are silent. When the trajectories from the Notch-On nuclei were partitioned into four states by vbSPT, the resulting populations had similar diffusions constants to those in Notch-Off conditions (Fig S1A, Table S1A), demonstrating that the overall properties were not substantially altered. However, the proportions mapping to each state differed. First, Notch activation led to a significant drop in the proportion of "free" CSL molecules (from 48% to 32%). Second there was an increase across the other populations, with the proportion

of bound molecules (D1 + D2 increasing from 25% to 31%, Fig 2A) and the proportion of exploratory, D3 molecules increasing (from 27.5% to 37%) (Fig 2A, Table S2A).

The increase in CSL binding and exploration was also reflected in the results from DDMap analysis, which yielded a larger proportion of sub-diffusive trajectories in Notch-On (Fig S1C). More striking were the changes in mean diffusion coefficients which decreased significantly in Notch-On conditions (Fig 2B). These changes are consistent with CSL becoming less freely diffusing and instead exhibiting more stable binding behaviour and more exploration of chromatin in Notch-On conditions.

CSL participates in two types of protein complexes. In Notch-Off conditions, CSL partners with Hairless to form a corepressor complex (Morel et al, 2001; Barolo et al, 2002; Yuan et al, 2016). In Notch-On conditions, a fraction of CSL molecules are recruited into a tripartite complex with NICD and the co-activator Mastermind (Petcherski & Kimble, 2000; Nam et al, 2006; Wilson & Kovall, 2006). It is likely that the change in CSL behaviours is related to the nature of the complexes present in the two conditions. We therefore set out to investigate whether the dynamics of Mastermind and Hairless in Notch-On conditions correlated with the different behaviours detected for CSL, by generating endogenously expressed Halo-tagged variants of the two proteins and tracking them with SMLM (Video 5 and Video 6). The trajectories were partitioned into four states using vbSPT, and in all cases the resulting populations had similar diffusions constants consistent with them participating in the same complexes (Fig S1A and B, Table S1A). The diffusion characteristics of Hairless were very similar in both Notch-Off and Notch-On conditions and resembled those of CSL in Notch-Off conditions, (Fig 2A and Table S2), with a relatively large proportion of D4-freely diffusing molecules (48%). Conversely, diffusion characteristics of Mastermind were more closely aligned with those of CSL in Notch-On conditions, with lower proportion of D4 molecules (25%) and a higher proportion of bound molecules (44%). Similar proportions of Mam also exhibited D3, "exploratory," behaviour (32% Mam versus 37% CSL). Thus, the Notch-induced changes in CSL behaviour are compatible with it being in a complex with Mam, albeit the proportions of Mam with "bound" D1/D2 behaviours (44%) were even greater than for Notch-On CSL (31%) suggesting some of the CSL remains in complexes with its other partner.

To investigate further the change in CSL behaviour in Notch-On conditions, we focused on the more dynamic molecules in the D3

---

black D3, yellow D4. **(D)** Average diffusion coefficients of four CSL populations assigned by vbSPT analysis of 50 ms movies, n = 7 nuclei. See Table S1 for mean values (±SD). For all boxplots, line indicates median, crosses represent mean, boxes 5–95 percentiles and whiskers upper and lower extrema. Data from these seven nuclei form the basis for analysis in (F, G, H, I, J, K). **(E)** Representative examples of trajectories from indicated vbSPT populations. **(F)** Distribution of displacements (*dis*) per population, measurements of *dis* the distance between two consecutive localisations in a trajectory as illustrated in the schematic. Trajectories from n = 7 nuclei in (D) (72,980 trajectories), were pooled. Displacement value dis = 0.035 ± 0.040 $\mu$m for D1, dis = 0.061 ± 0.050 $\mu$m for D2, dis = 0.123 ± 0.088 $\mu$m for D3 and dis = 0.271 ± 0.155 $\mu$m for D4. **(G)** Circular histograms show angle distributions per population, angle $\theta$ is measured between three consecutive localisations in a trajectory as illustrated in the schematics, pooling trajectories as in (F). The resultant vector, R, gives a measure of accumulation of angles around the mean value (higher R value shows more accumulation). Mean = 176°, 179°, 179°, 137° and R = 0.300, 0.331, 0.217, 0.008 for D1, D2, D3 and D4 respectively. **(H)** Pie chart showing average percentages of CSL molecules in each vbSPT population and diffusion coefficient. For D1–D4 respectively, mean values (±SD) of proportions are 5.7% ± 0.03%, 18.7% ± 0.03%, 27.5% ± 0.10% and 48.1% ± 0.10%. Mean values (±SD) of diffusion coefficients are 0.006 ± 0.001, 0.020 ± 0.002, 0.078 ± 0.008, and 0.512 ± 0.050 $\mu$m$^2$/s. D1–D4 populations are characterised as Bound1, Bound2, Exploratory and Free respectively, based on diffusion coefficients, displacement, and angle analyses. **(I)** Nuclear localisations of CSL single molecule trajectories from (C) coloured according to motion type assigned by DDMap analysis: Brownian (grey), sub-diffusion (blue) and super-diffusion (red). **(J)** Average diffusion coefficients of trajectories obtained from DDMap analysis of seven nuclei as in (D), grouped by motion type (means: 0.317 ± 0.066 $\mu$m$^2$/s for Brownian, 0.036 ± 0.007 $\mu$m$^2$/s for sub-diffusive). **(K)** Average proportion of trajectories per nucleus of each motion type obtained from DDMap analysis (means: 0.940 ± 0.016 for Brownian, 0.060 ± 0.016 for sub-diffusive).

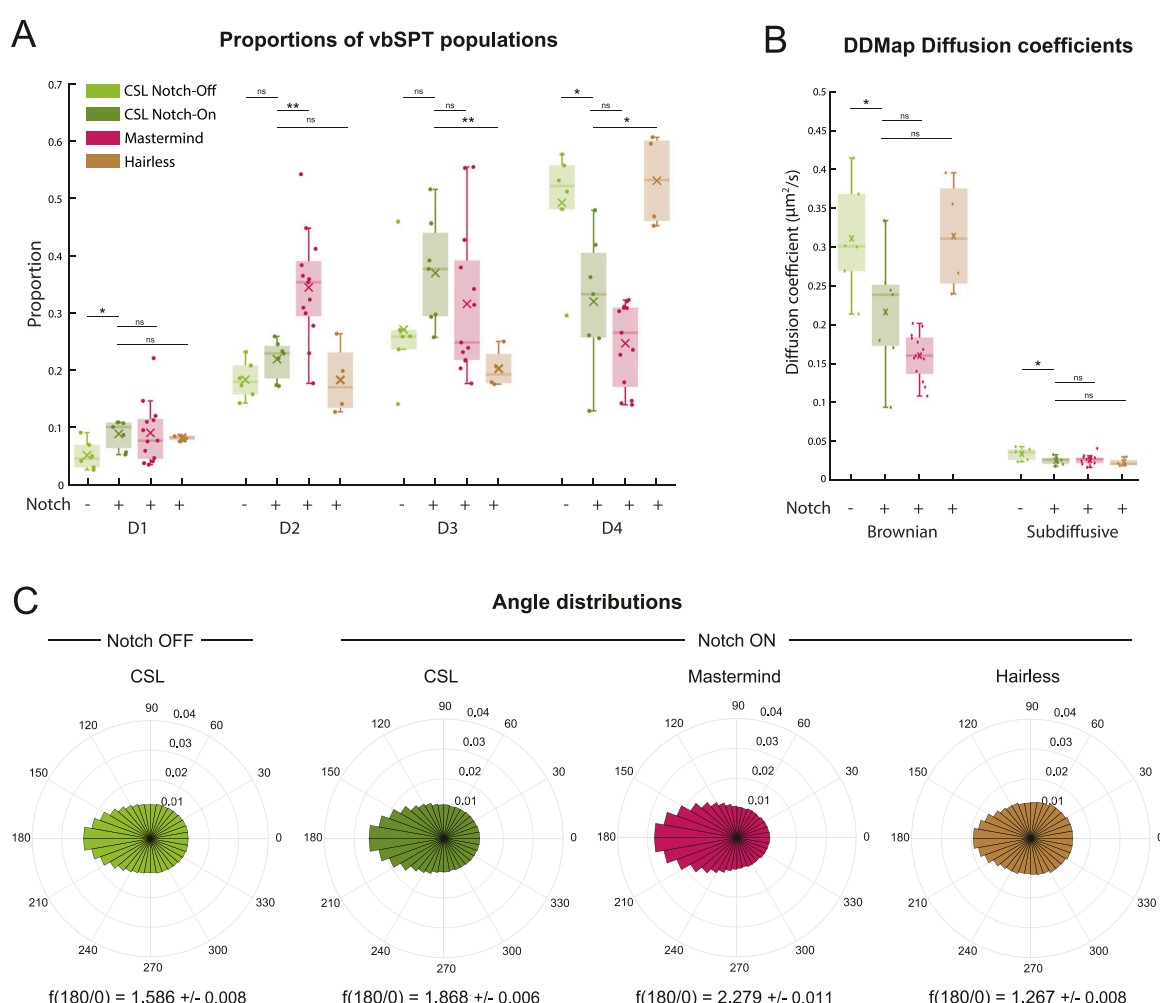

**Figure 2. Changes in the dynamics of CSL complexes in Notch active conditions.**
**(A)** Average proportions per nucleus of CSL (green), Mam (red) and Hairless (brown) molecules in each of four vbSPT populations in Notch-Off (−) or Notch-On (+) conditions as indicated (CSL Notch-Off n = 7 nuclei, 72,980 trajectories; CSL Notch-On n = 7 nuclei, 125,071 trajectories; Mam n = 13 nuclei, 66,753 trajectories; and Hairless n = 4 nuclei, 49,657 trajectories). For mean values see Table S2A. Significance was assessed with Mann-Whitney *U* tests, \*$P < 0.05$, \*\*$P < 0.01$, \*\*\*$P < 0.001$. See Table S3A for all *P*-values. **(B)** Average DDMap diffusion coefficients per nucleus in (A) of Brownian and sub-diffusive CSL (green), Mam (red) and Hairless (brown) molecules in Notch-Off (−) or Notch-On (+) conditions as indicated. For mean values see Table S1B. Significance was assessed with two-sample *t* tests, \*$P < 0.05$, \*\*$P < 0.01$, \*\*\*$P < 0.001$. For *P*-values see Table S3B. **(C)** Circular histograms of angles, calculated as in Fig 1G, from pooled D3 and D4 CSL trajectories (assigned by vbSPT) in Notch-Off and Notch-On, and from Mastermind and Hairless in Notch-On (nuclei from (A)). n = 220,711, 328,143, 142,760 and 121,396 angles respectively. The fold anisotropy metric f(180/0) ± SD is noted in each case (for more information on f(180/0) see the Materials and Methods section).

and D4 states and measured their anisotropy to ascertain whether, on average, their movement becomes more compact, which would indicate that some undergo a change in searching behaviour (Fig 2C). Indeed, there was an overall increase in the backwards anisotropy of CSL trajectories in Notch-On conditions, effectively reducing the size of the spaces explored. As with the overall dynamics, these properties were more similar to those of Mam than of Hairless—the latter exhibiting little backwards anisotropy.

These comparisons show that the behaviours of CSL correlate with those of its partners and shift from having more freely-diffusing characteristics in Notch-Off conditions, similar to Hairless behaviour, to having more bound and compact properties in Notch-On conditions, similar to Mastermind.

## Notch-On conditions promote recruitment and exploration at a target locus

The global changes in the properties of CSL could in theory increase the rate at which it binds to target sites. To investigate its movement and recruitment in relation to a specific genomic locus that it regulates, we focused on the *Enhancer of Split Complex* (*E(spl)-C*) a highly responsive locus containing 11 Notch-regulated genes (Delidakis & Artavanis-Tsakonas, 1992; Knust et al, 1992; Jennings et al, 1994; Bailey & Posakony, 1995; Lecourtois & Schweisguth, 1995; Lai et al, 2000). We have previously generated strains for live imaging *E(spl)-C*, introducing INT sequences into the 3′ end of the complex and combining this with fluorescently tagged ParB which recognizes INT, enabling us to

visualise the locus as a clear "band" in salivary gland nuclei (Fig 3A and B) (Gomez-Lamarca et al, 2018).

Taking a snap-shot of the *E(spl)-C* at the start and end of each movie, we compared the behaviour of CSL around this locus against behaviour elsewhere in the nucleus (Fig 3). From the density and trajectory maps of CSL, it was already evident that there was a significant enrichment of trajectories around *E(spl)-C* in Notch-On nuclei (Figs 3A, B, and D and S2A and B). To quantify the locus associated trajectories relative to those elsewhere in the nucleus, we set a distance threshold of 550 nm around *E(spl)-C* and every trajectory falling within that region was designated as "near" while all other trajectories were designated as "away" (Fig 3B). Firstly, we measured the molecule density (number of trajectories per unit area) near the locus and calculated its ratio to the nuclear density. The ratio was significantly higher in Notch-On conditions (Fig 3C), confirming that Notch activity promoted recruitment of CSL molecules to the target locus. As expected, Mam also became enriched near *E(spl)-C* in Notch-On condition, indeed to an even greater extent than CSL (Fig 3A and C). Conversely, no enrichment was detected for the co-repressor Hairless for which there was a relatively small number of trajectories present near the locus (Fig 3A and C).

To investigate which type of CSL behaviour was affected, we calculated the enrichment in proportion for each diffusive population near the locus in Notch-On conditions. Of the four populations only the CSL D1 "bound" population showed a robust enrichment in proportion, a pattern that was replicated by the Mam populations. This was accompanied by a relative decrease in the proportion of faster moving D4 molecules (Fig 3E and F). Despite not becoming substantially enriched at the locus (Fig 3A and C), Hairless also exhibited proportionately more bound-like behaviour, evidenced by increased proportions of D1, D2 molecules and reduced proportion of D4. The increase in binding near the locus for all three molecules was also captured by DDMap analysis, both through a slight enrichment of the sub-diffusive population and through a significant decrease in the diffusion coefficients of Brownian trajectories near this region (Figs 3G and S2B and C).

The increase in bound CSL molecules at *E(spl)-C* in Notch-On conditions is consistent with previous imaging and ChIP data (Krejčí & Bray, 2007; Castel et al, 2013; Wang et al, 2014; Gomez-Lamarca et al, 2018). The fact that there is proportionately more binding for all three molecules, including the corepressor Hairless, suggests that there is a change in the chromatin environment that facilitates the binding or trapping of CSL complexes in that region (Gomez-Lamarca et al, 2018; DeHaro-Arbona et al, 2023).

### Notch activity promotes local searching by CSL complexes

The increase in binding of CSL at *E(spl)-C* in Notch-On conditions led us to question whether there was a change in its local behaviour to favour its recruitment in the vicinity. For example, an increase in the anisotropy of trajectories, would reflect more compact diffusion around the locus that could aid recruitment. We therefore analysed the anisotropy of D3 and D4 trajectories near *E(spl)-C* in comparison to elsewhere in the nucleus. Based on the anisotropy metric f(180/0), there was a large increase in CSL backwards anisotropy near *E(spl)-C* (2.75 versus 1.85, Fig 4A). This demonstrates that the anisotropy is not uniform across the nucleus and suggests it is

associated with specific regions of the genome that are regulated by Notch activity such as E*(spl)-C*.

The anisotropy indicates there is extensive "back-tracking" and implies that the effective size of the space CSL explores is reduced, by local trapping or transient binding, which would favour its on-rate at the target locus (Slutsky & Mirny, 2004; Kapanidis et al, 2018; Woringer & Darzacq, 2018; Hansen et al, 2020; Darzacq & Tjian, 2022). If this model is correct, the anisotropy should peak for displacement lengths corresponding to the size of regions where the motion of the molecules is constrained. We therefore asked how anisotropy changed with displacement length, keeping our analysis within a displacement range of 50–350 nm. This provided accuracy by excluding displacements close to/lower than our localisation error (20 nm) (see the Materials and Methods section) and by ensuring sufficient numbers of angles for the longest displacements in the range. First it was evident that the pattern of CSL anisotropy differed near the locus compared to away. Notably, near the locus CSL, acquired a peak of anisotropy between 100–150 nm, suggesting that there is a tendency for CSL molecules to become trapped in zones of that size (Fig 4D). Combined with the fact that anisotropy dropped for longer displacements, it implies a form of local exploration near the locus (Slutsky & Mirny, 2004; Kapanidis et al, 2018; Woringer & Darzacq, 2018; Hansen et al, 2020; Darzacq & Tjian, 2022). A similar behaviour was observed for the co-activator Mam, whose anisotropy also displayed a peak at the 100–150 nm range (Fig 4B and E). In contrast, Hairless showed no similar peak in anisotropy near the locus for any specific displacements (Fig 4C and F) arguing that the corepressor complex lacks the features that favour local searching.

Overall, our results show that Notch activity brings about changes in the behaviour of CSL at target loci. The properties it acquires, of increased binding and a more restricted compact diffusion, resemble those of Mam, consistent with the two forming a complex. Their anisotropic diffusion close to the target locus implies that they engage in local exploration (Izeddin et al, 2014; Hansen et al, 2020; Mazzocca et al, 2021) that could aid their recruitment.

### Clustering analysis reveals higher backwards anisotropy near clusters of bound CSL

To investigate whether the searching behaviour observed at *E(spl)-C* is a more general feature of Notch-On conditions, we asked whether similar anisotropy was present at other genomic locations where CSL exhibits "bound" behaviours. First, we identified clusters of bound CSL (D1 and D2) molecules in each of our datasets (excluding those at *E(spl)-C*) and then analysed the anisotropy of the more motile populations in relation to those clusters.

A method to detect features, regions of high point densities, in spatial data that has a high degree of "clutter" (Byers & Raftery 1998, see the Materials and Methods section) was used to find clusters of slow trajectories. The bound (D1 and D2) trajectories were described by their average position (barycentre) and then these positions were modelled as a superposition of two spatial Poisson processes: a high intensity Poisson process describing clustered trajectories and a low intensity process describing non-clustered trajectories, ("clutter"). The trajectories belonging to these two classes were distinguished based on their kth nearest neighbour distance to other trajectories. Thus, the distance to the nearest neighbour was

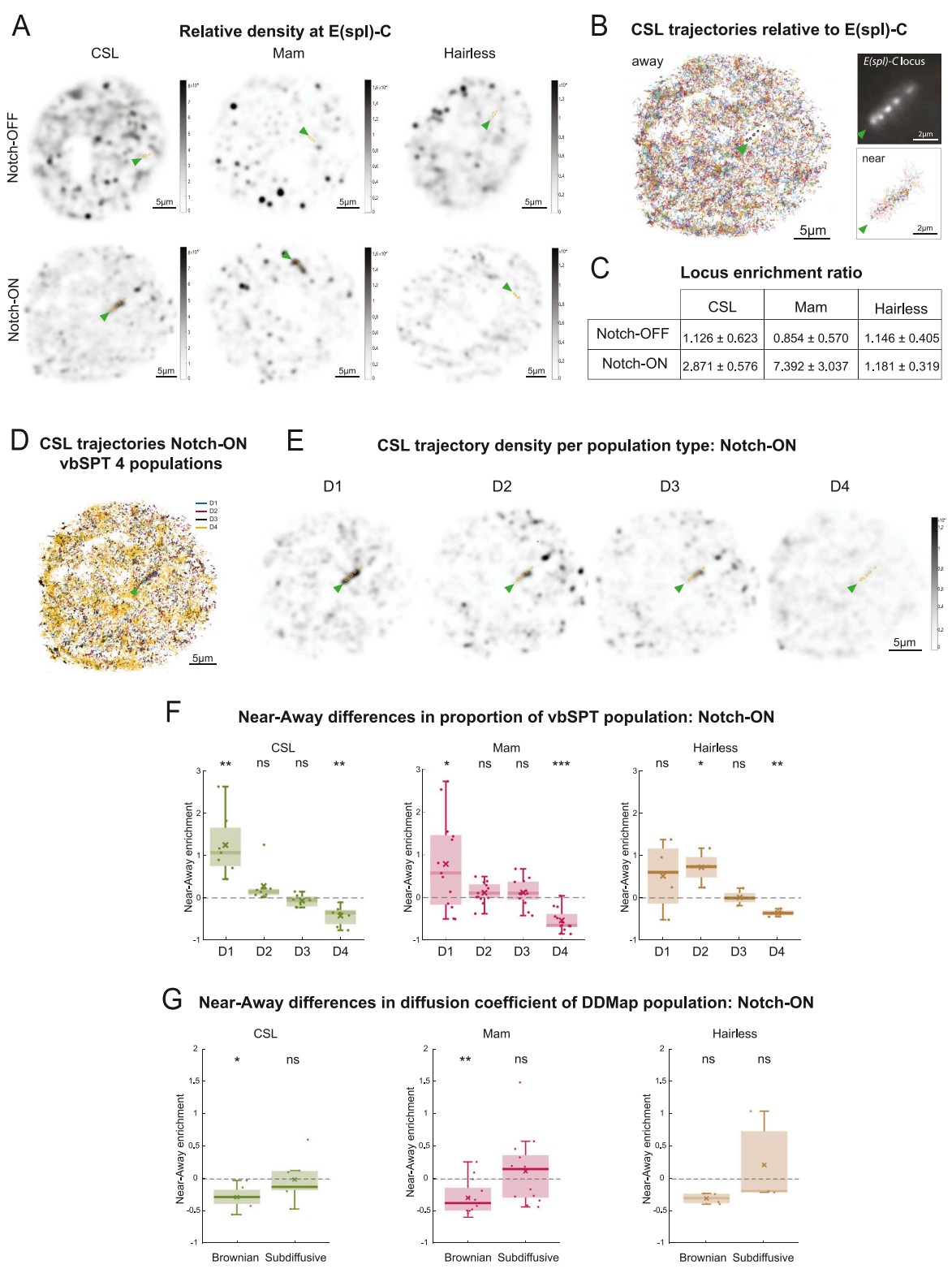

**Figure 3. Recruitment and change in behaviour of CSL complexes near *E(spl)-C* in Notch active conditions.**
**(A)** Density maps of CSL, Mam and Hairless trajectories from representative Notch-off and Notch-On nuclei. Colourbar is in units of number of trajectories per $nm^2$. Arrowhead and yellow outline indicate *E(spl)-C* locus. **(B)** Segregating trajectories relative to *E(spl)-C* (detected by Int/ParB-GFP, arrowhead and upper inset) in a representative Notch-On nucleus. Trajectories localised within a 550 nm distance area around *E(spl)-C* locus were classified as "near" (lower inset). Remaining trajectories in the nucleus were classified as "away" (main image). Arrowhead indicates location of *E(spl)-C* locus. **(C)** Mean locus enrichment ratio (±SD) is shown (see the Materials and Methods section). A ratio >1 indicates molecule density near *E(spl)-C* is higher than overall nuclear density. For Notch-Off conditions, CSL n = 7 nuclei, 72,980

calculated for all molecules and they were then divided into two groups: one which had small distance to its kth nearest neighbour (clustered) and one which had large distance to kth nearest neighbour (non-clustered). The analysis remained robust over a range of k values and we focused on k = 11, which identified large and dense clusters, making these regions plausible candidates for other target enhancers (Figs 5 and S3A).

Based on this analysis, clustering of CSL bound molecules was significantly more common in Notch-On nuclei (Fig 5A, C, and E), where the proportion of clustered bound molecules was 1.7-fold higher than in Notch-Off. To analyse the behaviour of molecules close to the clusters we set a 550 nm search radius and captured all D3 and D4 localisations falling within that region (Fig 5B and D). These constituted what we defined as "near-cluster" jumps, while all other jumps were "away-cluster," and we calculated what proportion of "near" jumps were anisotropic (see the Materials and Methods section). As a control we randomly shifted the position of the clusters and repeated the anisotropy analysis for localisations near the "fake" clusters (see the Materials and Methods section). Our results showed a significantly higher anisotropy in the diffusion properties of molecules near the bone fide CSL clusters compared to those located away from clusters in Notch-On conditions (Fig 5F). Unexpectedly, this difference was also detected in Notch-Off conditions, albeit there were many fewer CSL clusters (Figs 5E and S3B). These results suggest that the change in anisotropy is linked to the clustering of bound complexes rather than to a specific property of the activation complexes themselves. One likely explanation is that the bound clusters are associated with regions of more accessible chromatin, which would provide an environment that favoured searching, giving rise to more anisotropic behaviours. While increased anisotropy was observed near bound clusters regardless of Notch activity status, when comparing the proportion of anisotropic displacements that were located near bound clusters, we found this to be much higher in Notch-On conditions (42.8% versus 13.2%) (Fig 5G). If our bound cluster/open chromatin correlation hypothesis is correct, the accumulation of anisotropic behaviour near these clusters in Notch-On could indicate a mechanism through which local searching in regions of open chromatin enables these activation complexes to efficiently find their target sites.

## Discussion

In summary, tracking the behaviour of CSL molecules in real time reveals that they acquire more exploratory behaviours in Notch active conditions. Characterised by restricted and anisotropic diffusion, these behaviours are most evident close to a target gene locus and could aid recruitment and binding at regulated enhancers (Fig 5H). Anisotropy is thought to originate from a combination of two mechanisms. One probable cause is reattachment to the DNA. When a protein dissociates from a site where it is bound, it is more likely to re-attach to the same region rather than binding elsewhere. In the time-scale of SPT experiments (milliseconds) the protein will likely step back following a forward jump. The second mechanism involves protein trapping. If a protein engages in interactions with other factors, it will be reflected back from the boundary of protein rich domains, leading to an increased probability of backward motion. The range of marked anisotropy is indicative of the size of the zone where the protein is trapped (Hansen et al, 2020). The properties are thus consistent with Notch activity promoting formation of a local protein rich hub at target enhancers that involves a combination of altered chromatin accessibility, increasing the probability of DNA-protein interactions, and multivalent protein interactions, retaining the proteins in proximity (Fig 5H).

Similar search strategy has been observed for other transcription factors. For example, p53 alternates between fast and more compact diffusion, the latter being characterised by anisotropic diffusion at short spatial scales similar to those we observe for CSL. Both factors thus appear to repeatedly sample domains of ~100–150 nm that frequently co-localize with clusters of bound molecules, suggesting that they locally scan the genome to identify their target sites (Mazzocca et al, 2023). Also referred to as guided exploration, these behaviours are emerging as a common mechanism to accelerate the search process of nuclear proteins. For example, the chromatin factor CTCF and the Polycomb subunit CBX also display compact diffusion profiles similar to the ones measured for p53 and CSL. Both CTCF and CBX were shown to form visible clusters or condensates (Hansen et al, 2020; Kent et al, 2020) that could explain the trapping zones observed and may be similar to the zones of high CSL density that we have observed in Notch-On nuclei by light microscopy (Gomez-Lamarca et al, 2018; DeHaro-Arbona et al, 2023). Our results, along with these other examples, suggest a general mechanism where local changes in protein composition and chromatin accessibility, promote local searching to enable efficient recruitment of transcription complexes to their target sites. In the case of CSL, Notch-On conditions, and its association with Mam, are required for it to acquire these properties.

While the acquisition of increased local searching by CSL in Notch-On correlates with the behaviour of the co-activator Mam, we also detected more bound and less mobile Hairless molecules in the vicinity of E(spl)-C in Notch-On nuclei. As Hairless is a co-

trajectories, Mam n = 5 nuclei, 24,322 trajectories and Hairless n = 5 nuclei, 66,837 trajectories. For Notch-On conditions, CSL n = 7 nuclei, 125,071 trajectories, Mam n = 13 nuclei, 66,753 trajectories and Hairless n = 4 nuclei, 49,657 trajectories. **(D)** Single molecule trajectories from a representative CSL Single Molecule Localisation Microscopy experiment in a Notch-On nucleus, plotted at their respective localisation positions, colour coded according to vbSPT diffusion state (Blue D1, Red D2, Black D3, Yellow D4). Green arrowhead indicates location of *E(spl)-C* locus. **(E)** Density map of each CSL population (D1–D4) from (C) relative to *E(spl)-C* locus (arrowhead and yellow outline). Colour-bar is in units of number of trajectories per nm². **(F)** Enrichment of vbSPT populations near *E(spl)-C* for CSL, Mam and Hairless in Notch-On conditions (nuclei as in (B)). Values plotted were calculated as (Proportion near- Proportion away)/Proportion away, hence a value greater than 0 represents an enrichment of the respective population near the target locus. Significance was assessed using two-sample *t* tests, *$P < 0.05$, **$P < 0.01$, ***$P < 0.001$ and see Table S3D for all *P*-values. **(G)** Relative diffusion coefficients of Brownian and sub-diffusive trajectories near *E(spl)-C* versus away for CSL in Notch-On conditions (nuclei as in (B)). Values plotted were calculated as (Diffusion coefficient near- Diffusion coefficient away)/Diffusion coefficient away, hence a value smaller than 0 indicates a decrease in the diffusion coefficient near the locus. Significance was assessed using Wilcoxon signed rank-sum tests, * for $P < 0.05$, ** for $P < 0.01$, *** for $P < 0.001$ and see Table S3E for all *P*-values.

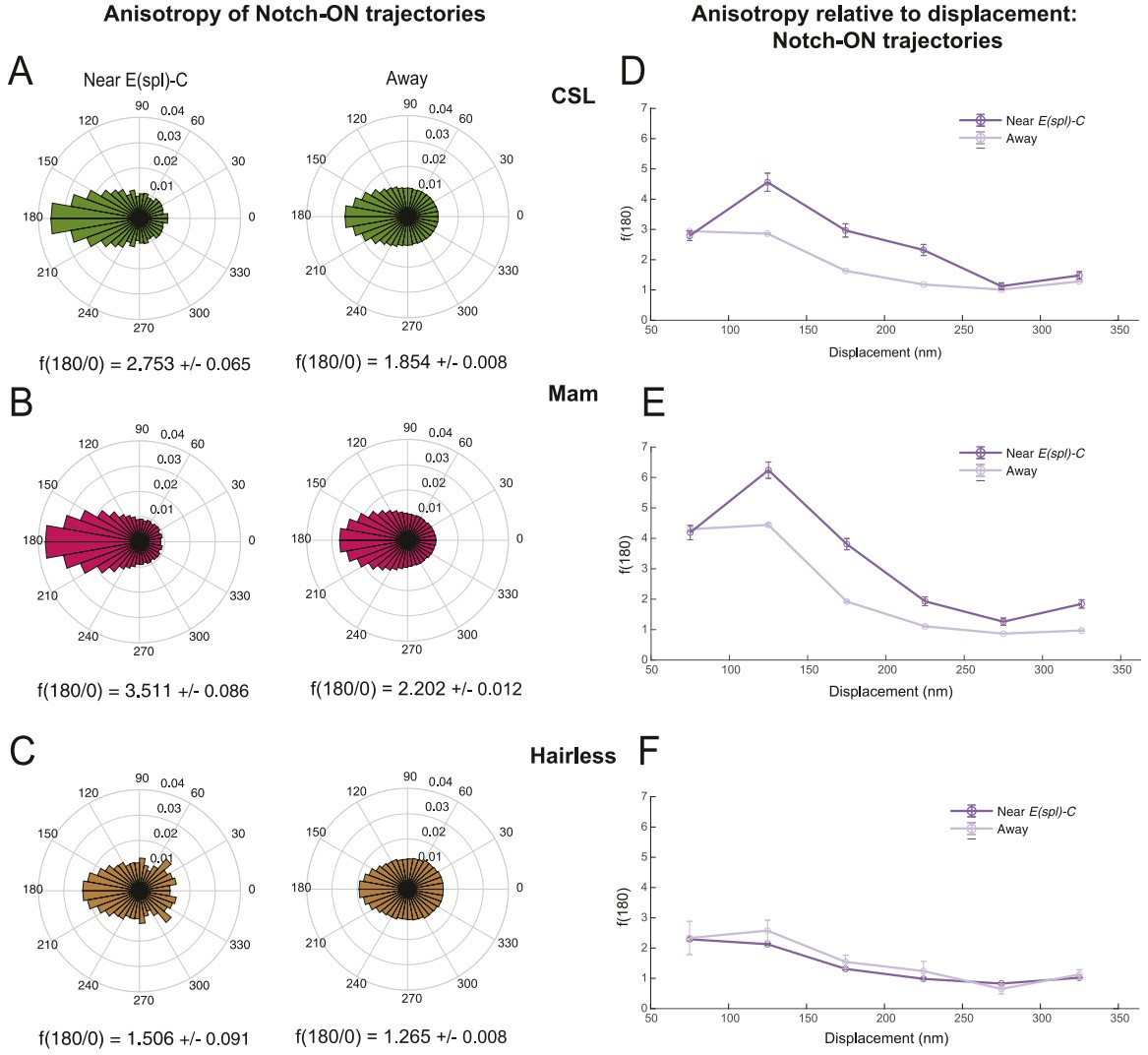

**Figure 4. Increased Searching behaviour of CSL complexes at target locus in Notch-On conditions.**
**(A, B, C)** Circular histograms of angles calculated from pooled D3 and D4 trajectories (assigned by vbSPT) for CSL (A), Mam (B) and Hairless (C) in Notch-On nuclei near *E(spl)-C* locus and away from it (near/away segregation as shown in Fig 3B). CSL n = 6,158 angles near, n = 321,985 angles away; Mam n = 10,371 angles near, n = 132,389 angles away; and Hairless n = 945 angles near, n = 120,451 angles away. The value of f(180/0) ± SD is also given for each distribution. **(D, E, F)** Anisotropy [f(180/0)] relative to molecule displacement (nm) for D3 and D4 trajectories of CSL (D), Mam (E) and Hairless (F) near *E(spl)-C* locus and away in Notch-On conditions (near/away segregation as shown in Fig 3B). Error bars show the SD from bootstrapping with 50 iterations (for more information see the Materials and Methods section).

repressor, at first glance it seems surprising that there is a local increase in its association with chromatin at a positively regulated gene locus. A likely explanation is that the activation complex brings about a local change in chromatin accessibility which would make the region more favourable to binding of CSL complexed with Hairless as well – a concept analogous to assisted loading (Voss et al, 2011; Gomez-Lamarca et al, 2018). Presence of co-repressor could enable a more rapid shut down when signalling levels decrease and/or function as an amplitude rheostat (facilitated repression: Zhu et al, 2015; Gomez-Lamarca et al, 2018). The limited and local derepression of target genes that occurs in cells with reduced Hairless function are very compatible with these models (Chan et al, 2017).

Another notable feature of CSL in our experiments is that a relatively small proportion of the molecules appear to be stably associated with chromatin, based on the proportions in states D1 and D2 states. This differs from the characteristics shown by pioneer factors such as the zinc-finger GAGA associated factor, most of which is chromatin bound, with a stable-binding fraction showing nucleosome-like confinement (Tang et al, 2022). Other pioneer factors such as Zelda exhibit more dynamic binding but still have a substantially greater proportion of bound/chromatin associated molecules than we observe with CSL (Mir et al, 2018). Whether this difference is due to the pioneers having longer residence time than CSL or to their ability to access a greater spectrum of chromatin, including regions that would be inaccessible to other types of

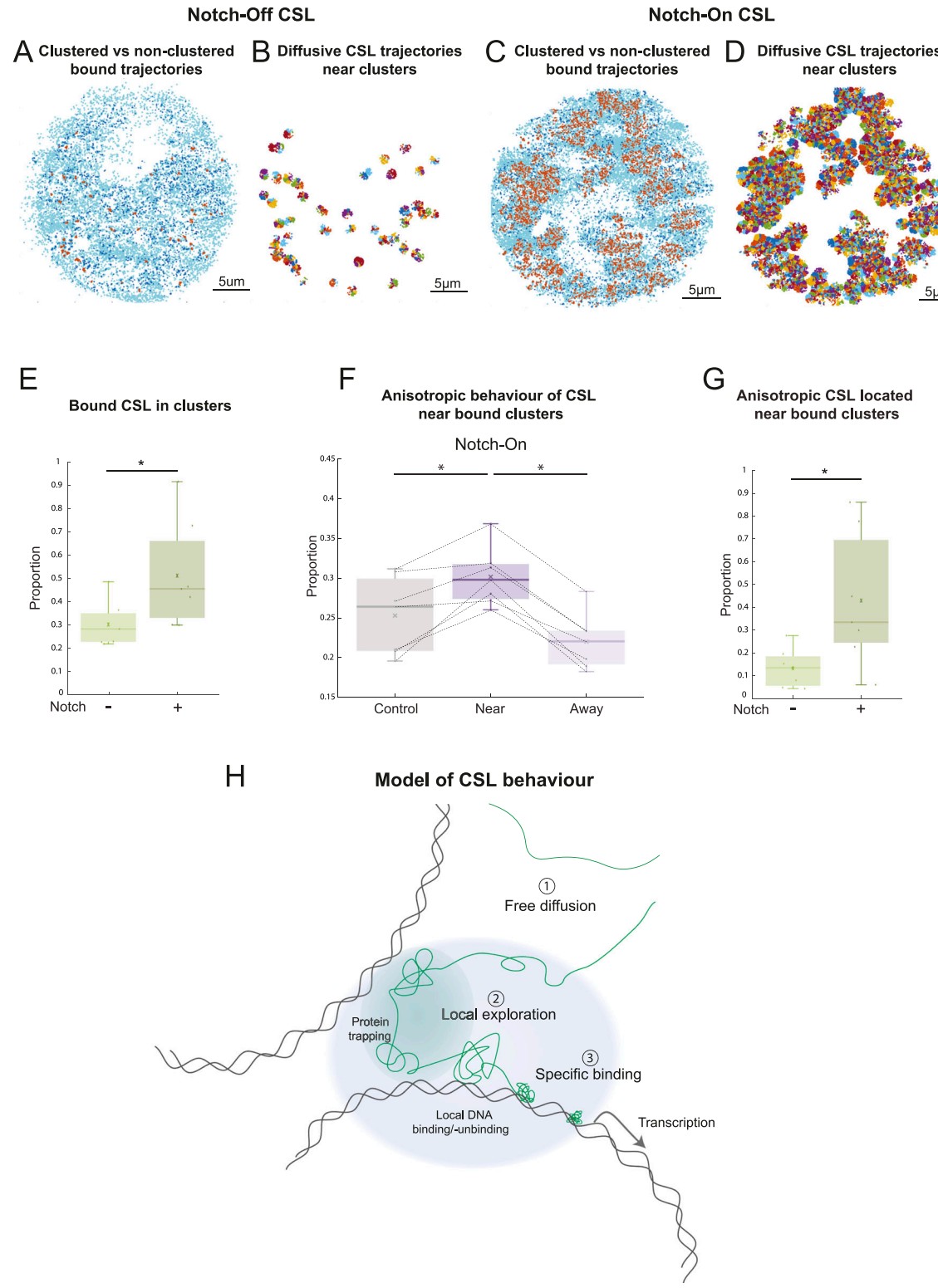

**Figure 5. Clustering analysis reveals higher backwards anisotropy near clusters of bound CSL.**
**(A, C)** Clustered bound trajectories identified using a superposition of Poisson process models from representative examples of CSL Notch-Off (A) and Notch-On (C) nuclei. Each dot represents the centre of a trajectory. Orange, clustered bound trajectories; dark blue, bound trajectories identified as non-clustered; light blue, diffusive trajectories (D3 and D4). **(B, D)** CSL diffusive jumps (D3 and D4) in nuclei shown in (A, C), identified as "near-cluster" using a 550 nm distance threshold around each clustered bound trajectory. **(E)** Proportion of bound trajectories that were identified as clustered for CSL Notch-Off (0.301 ± 0.096) and Notch-On (0.511 ± 0.228) nuclei. A two-sample $t$ test was performed, $P$ = 0.045. **(F)** Proportion of near and away diffusive jumps (D3 and D4) that were anisotropic in CSL Notch-On nuclei. Mean values: 0.252 ±

transcription factors like CSL because they have high nucleosome densities, remains to be established.

One limitation of our study is that the *Drosophila* salivary gland is unusual, having multiple genomic copies in its polytene chromosomes. This organisation facilitates the imaging and monitoring of changes with respect to a target gene but may have some unusual properties. For example, there may be more extensive local hopping between the co-aligned DNA strands that could favour more exploratory behaviour. We note also that while we are following endogenous levels of the nuclear factors, Notch activity is provided ectopically. However, our observations are consistent with the increased CSL recruitment detected by ChIP in a range of cell types (Krejčí & Bray, 2007; Castel et al, 2013; Wang et al, 2014), making it likely that they reflect general properties of these transcription complexes.

# Materials and Methods

### Experimental animals

*Drosophila melanogaster* flies, as indicated, were maintained at room temperature (~20°C) using standard cornmeal food (Glucose 76 g/liter, Cornmeal flour 69 g/liter, Methylparaben 2.5 ml/liter, Agar 4.5 g/liter and Yeast 15 g/liter.) Experimental crosses, from which larvae were collected for dissections, were kept at 25°C.

### Fly stocks and crosses

Notch activity was provided by combining *UAS-NΔECD* (Fortini et al, 1993; Rebay et al, 1993) with *1151-Gal4* (FBti0007229, Gomez-Lamarca et al, 2018), the control *UAS-LacZ* (FBti0018252) was used in Notch-Off glands. The indicated Halo-tagged lines, as described below, were present in each condition and the *E(spl)-C* locus was labelled with an INT insertion detected by ParB1-GFP (Gomez-Lamarca et al, 2018). Full genotypes are provided in Table S4.

### Generating Halo-tagged protein lines

To generate lines expressing endogenous levels of Su(H)::Halo and Hairless::Halo, we modified the AttB plasmids containing genomic rescue constructs Su(H)-eGFP and Hairless:eGFP, (Gomez-Lamarca et al, 2018). Briefly, eGFP coding sequence was replaced by HaloTag (pFN23K-Halo plasmid, given by St Jonhston lab, G2861; Promega), amplified using primers CACCTAGGATGGCAGAAATCGGTACTGGCTTTCCATTCGACC and CTACGCGTTGCCGGAAATCTCGAGCGTGG for Su(H)::Halo, and primers TTACAGATCTCTGAAATCGGTACTGGCTTTCC and TTTCTAGAGACAGCCGGAAATCTCGAGCGTGG for Hairless::Halo. Plasmids and HaloTag PCR

products were digested using AvrII (R0174S; New England Biolabs) and MluI (R0198S; NEB) for Su(H)::Halo, or BglII (R0144L; NEB) and XbaI (R0145S; NEB) for Hairless::Halo. The resulting AttB plasmids were injected into a strain containing phiC31 integrase and AttP site in position 86F8 in chromosome 3 (Bloomington 24749; Su(H)) or position 51D in chromosome 2 (Bloomington stock 24483; Hairless) to generate transgenic Su(H)::Halo or Hairless::Halo flies.

Mam::Halo flies were generated using CRISPR/Cas9 genome engineering (flycrispr.org) to insert the coding sequence of HaloTag into the endogenous Mam gene. Briefly, a plasmid containing homology arms flanking the starting codon of Mam, HaloTag, SV40 and the PiggyBac 3xPax3-dsRED cassette from pHD-ScarlessDsRed (flycrispr.org) was synthesized by NBS Biologicals. Transformants were obtained by co-injecting this plasmid with a pCFD3-dU6:3gRNA plasmid (#49410; Addgene) expressing the gRNA GACGCATTTATG-GATGCGGG and screening for 3xPax3dsRED. The 3xPax3-dsRED cassette was excised by crossing with αTub84B-PiggyBac flies (#32070; BDSC). Maps of the homology and gRNA plasmids and final genomic sequence can be found at https://benchling.com/bray_lab/f_/bRaeXvzx-mamhalo-endogenous-tag/.

### Salivary gland cultures and Halo ligand treatments

Salivary glands were isolated from third instar larvae grown at 25°. Dissections and mounting in observation chambers were performed as described (Gomez-Lamarca et al, 2018) except for the additional following steps to sparse label with Halo ligand. Glands were incubated for 15 min in TMR Halo ligand (G825A; Promega) diluted in Dissecting medium (Gomez-Lamarca et al, 2018). Concentration of ligand was first adjusted in each case by serial dilutions to reach single molecule resolution which corresponded to a concentration of 10 nM for CSL-Halo and Hairless-Halo, of 50 nM for Mam-Halo and of 0.01–0.02 nM for H2AV-Halo. After incubation the glands were washed 3 × 10 min in Dissecting medium and briefly rinsed in PBS(1X) (#70011044; Thermo Fisher Scientific) before mounting.

### SMLM

The custom build microscope and localisation errors were as described in Gomez-Lamarca et al (2018). Samples were continuously illuminated during imaging with a 561 nm excitation wavelength laser to excite the Halo ligand emitting at 585 and a 488 nm excitation wavelength laser to excite the GFP-labelled Locus Tag emitting at 510 nm. Laser power used for imaging with these lasers was ~500 and 30 W/cm$^2$ respectively. For each data set, the region imaged was focused around the nucleus and consisted of a square of ~40 μm × 40 μm dimensions. The pixel size of acquired movies

0.049 Control, 0.301 ± 0.037 Near, 0.220 ± 0.035 Away. Lines connect data from the same experiment (see the Results and Materials and Methods section for details on control analysis). Wilcoxon signed-rank tests were performed, $P = 0.016$ for both Control versus Near and Near versus Away. **(G)** Proportion of all anisotropic diffusive jumps (D3 and D4) that were identified as "near-cluster" for CSL Notch-Off (0.132 ± 0.084) and Notch-On (0.428 ± 0.291). A two-sample $t$ test was performed, $P = 0.036$. **(H)** Schematic illustrating trajectories of CSL complexes (green lines) in relation to a regulated gene locus. Away from chromatin, CSL complexes are freely diffusing (1). In Notch-On conditions, CSL acquires more exploratory behaviour (2) likely arising from protein trapping in confined regions (shaded green region) and local binding-unbinding to more accessible DNA. Together these result in a zone of localised enrichment (light blue shading) and favour an increase in specific-binding (3) at regulated genes, promoting transcription.

was 110 nm. Between 4–13 nuclei were imaged for each condition each with 50 ms exposure time for 3.3–6.7 min. For the experiments with 10 ms exposure time, three nuclei were imaged for 2.8–3.3 min.

### Trajectory analysis basic pipeline

Following movie acquisition, single molecules were localised with a Gaussian fitting-based approach (Ovesný et al, 2014) which allowed the enhancement of localisation precision to subpixel resolutions of ~20 nm (Gomez-Lamarca et al, 2018). A multiple hypothesis tracking algorithm was employed for tracking of the molecules, allowing no detection gaps within tracks, using an Icy-plugin based on Chenouard et al (2013). Trajectories consisting of at least four time points were then analysed using two different methods. The first method was used to partition trajectories into distinct diffusive states and was based on variational Bayes analysis (vbSPT) (Persson et al, 2013), where the trajectories were divided in sub-trajectories characterised by a Brownian diffusion coefficient. Setting a ceiling of four populations (with the exception of control analysis with H2AV where the ceiling was set to two), the finite number of diffusion coefficients allowed per experiment was chosen via Bayesian model selection, by maximising the score $F = \ln p(x|N)$ where $p(x|N)$ is the probability of the data conditioned on the number of states (Persson et al, 2013). For all experiments, the model selected was that of four states. To account for the bias towards slow jumps due to defocalisation (fast molecules leaving the focal plane much quicker than slow ones, Kues & Kubitscheck, 2002; Mazza et al, 2012; Hansen et al, 2018), any analysis involving population proportions was carried out using the whole trajectories, assigning to each trajectory the mode state of its jumps.

The second analysis method was used to examine a wider class of motion types and was based on a Dense Mapping approach (DDMap) (Salomon et al, 2020). As this method uses Langevin equations (Lemons & Gythiel, 1997) and therefore takes into account both diffusion and drift, it enabled us to consider Brownian, sub-diffusion and directed motion (Saxton, 1993). Each trajectory was classified as one of these motion types using a nonparametric three-decision statistical test and a unique diffusion coefficient was calculated for it. DDMap scripts were adapted so that non-spatially averaged diffusion coefficients were calculated. For both trajectory analysis methods, the target locus in each nucleus was detected using image thresholding with Otsu's method and a 2-D Gaussian smoothing kernel was used for filtering.

### Angle and anisotropy analyses

Standard MATLAB procedures were used for angle calculation, as shown in Fig 1G. Angular statistics, including the calculation of the resultant vector, R, were carried out using the MATLAB package CircStat (Berens, 2009). For anisotropy analyses, the fold anisotropy metric f(180/0) was defined and calculated as how many-fold more likely a molecule is to make a step backwards compared to a step forward,

$$f(180/0) = P\left(\frac{180° \pm 30°}{0° \pm 30°}\right)$$

This definition, and scripts required for this analysis were adapted from Hansen et al (2020). Using only D3 and D4 jumps (as identified by vbSPT analysis) for anisotropy analyses guaranteed the displacements considered to be well above our localisation error (Gomez-Lamarca et al, 2018), ensuring the accuracy of angle calculations. Calculations of f(180/0) against mean displacement were carried out for 50 subsamples using 50% of the data (with replacement). Error bars on Fig 4D–F show SD from these sub-samplings.

### Molecule density analysis

Molecule density maps showing the spatial distribution of molecule trajectories were generated using kernel smoothing in MATLAB, with 4 pixel (440 nm) bandwidth (Bowman et al, 1997). The units for all density maps are number of trajectories per 0.0001 $\mu m^2$. The design of the maps was carried out using the MATLAB package ColorBrewer (Stephen23, 2023). Values shown in Fig 3C as "Locus enrichment ratio" represent a fold increase of near-locus density compared to nuclear density and were calculated as follows:

$$\frac{No.\ of\ trajectories\ near\ locus}{Total\ no.\ of\ trajectories\ in\ nucleus} \times \frac{Nucleus\ area}{Locus\ area}$$

Locus and nucleus areas were calculated with standard MATLAB procedures, using the convex hull of localisations and masking.

### Clustering

Clustering of bound molecules, described by their barycentre (average position), was performed in R using *nnclean*. It is based on a superposition of random (Poisson) processes model (Byers & Raftery, 1998) and uses the distance of each point to its kth nearest neighbour to distinguish molecules belonging to two Poisson processes: a high intensity ($\lambda_1$) Poisson process for clustered trajectories and a low intensity ($\lambda_2$) Poisson process describing non-clustered trajectories (Fig S3A shows the histogram of the kNN distances for one data set). The analytical expression of the kth NN distance distribution of a random Poisson process is known to be a transformed gamma distribution,

$$D_k^2 \sim \Gamma(k, \lambda\pi)$$

and the analytical expression of the mixture of two kNN distance distributions becomes:

$$D_k \sim p\Gamma^{1/2}(k, \lambda_1\pi) + (1-p)\Gamma^{1/2}(k, \lambda_2\pi)$$

Through an expectation-maximisation algorithm, the parameters of the mixture distribution are estimated together with the "missing" data—the indicator if a point belongs to a cluster or not. Results of the clustering are visualised in Fig 5A and C (clutter is shown in dark blue, and clusters in orange). The analysis remained robust over a range of k values.

For anisotropy proportions shown in Figs 5F and G and S3B, a jump of a D3 or D4 trajectory was regarded as anisotropic if the angle it formed with its successive jump was within the range 180° ± 30°.

For each experiment, control analysis was performed in MATLAB by first identifying a region of interest using the convex hull of diffusive (D3 and D4) jumps. The localisation of bound molecules identified as "clustered" was randomly shifted to a different place in the nucleus, in this way the same "cluster shape" was retained for the control analysis and all bound clusters were translated by the same vector. If more than 80% of these molecules remained inside the region of interest after translation, anisotropy analysis was carried out for the diffusive molecules near and away the shifted clusters. The process was repeated 100 times for each movie and mean values were calculated.

### Boxplots and statistical tests

For all boxplots, lines across represent the median, crosses represent the mean, boxes 5–95 percentiles and whiskers upper and lower extrema.

For statistical tests involving only one sample, one sample *t* tests (for normal samples) and Wilcoxon rank-sum tests (for not normal samples) were performed. For statistical tests involving two samples, two sample *t* tests (if samples were normal) and Mann-Whitney *U* tests (if samples were not normal) were performed. For paired data (clustering analysis in Fig 5), paired *t* tests (if samples were normal) and Wilcoxon signed rank tests (if samples were not normal) were performed. Normality of the samples was assessed using Q-Q plots and Shapiro-Wilk tests. Where two samples were compared, equality of variance was also assessed with Bartlett's test (if samples were normal) and Levene's test (if samples were not normal). In all cases significance was presented as follows: * for $P < 0.05$, ** for $P < 0.01$, *** for $P < 0.001$. All statistical tests were performed in R.

## Data Availability

Scripts for SMT trajectory analysis are available in GitLab: https://gitlab.developers.cam.ac.uk/cr607/smt_trajectory_analysis. Raw data from SMLM experiments are available on FigShare: https://figshare.com/projects/Changes_in_searching_behaviour_of_CSL_transcription_complexes_in_Notch_active_conditions/187665.

## Supplementary Information

## Acknowledgements

We thank Kevin O'Holleran, Martin Lenz and Cambridge Advanced Imaging Centre for their advice and help with the imaging, Kat Millen for embryo injections to generate the Halo-tagged proteins, all members of the Bray lab for helpful discussions. The work was funded by a Wellcome Trust Investigator Award (212207/Z/18) to S Bray and by an ESPRC Fellowship (EP/R025398/1 and EP/Y008715/1) to L Muresan. C Roussos and S Baloul were supported by studentships from Wolfson College-Dept of Physiology Development and Neuroscience-School of Biological Sciences (University of Cambridge).

### Author Contributions

S Baloul: conceptualization, data curation, formal analysis, validation, investigation, visualization, methodology, and writing—original draft, review, and editing.
C Roussos: conceptualization, data curation, software, formal analysis, validation, investigation, visualization, methodology, and writing—original draft, review, and editing.
M Gomez-Lamarca: conceptualization, resources, supervision, methodology, and writing—original draft, review, and editing.
L Muresan: conceptualization, data curation, software, formal analysis, supervision, methodology, and writing—original draft, review, and editing.
S Bray: conceptualization, resources, supervision, funding acquisition, visualization, project administration, and writing—original draft, review, and editing.

### Conflict of Interest Statement

The authors declare that they have no conflict of interest.

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
