## [Reviewer comments · Life Science Alliance]

Life Science Alliance

Changes in searching behaviour of CSL transcription complexes in Notch active conditions

Sarah Baloul, Charalambos Roussos, María Jesús Gómez Lamarca, Leila Muresan, and Sarah Bray

DOI: <https://doi.org/10.26508/lsa.202302336>

Corresponding author(s): Sarah Bray, University of Cambridge and Leila Muresan, University of Cambridge

Review Timeline:	Submission Date:	2023-08-24
	Editorial Decision:	2023-09-25
	Revision Received:	2023-11-05
	Editorial Decision:	2023-11-27
	Revision Received:	2023-12-03
	Accepted:	2023-12-04

Transaction Report:

September 25, 2023

Re: Life Science Alliance manuscript #LSA-2023-02336-T

Prof. Sarah J. Bray
University of Cambridge
Dept. of Physiology, Development and Neuroscience
University of Cambridge
Downing Street
Cambridge, United Kingdom

Dear Dr. Bray,

Thank you for submitting your manuscript entitled "Changes in searching behaviour of CSL transcription complexes in Notch active conditions" to Life Science Alliance. The manuscript was assessed by expert reviewers, whose comments are appended to this letter. We invite you to submit a revised manuscript addressing the Reviewer comments.

Thank you for this interesting contribution to Life Science Alliance. We are looking forward to receiving your revised manuscript.

Sincerely,

B. MANUSCRIPT ORGANIZATION AND FORMATTING:

Reviewer #1 (Comments to the Authors (Required)):

The work in this manuscript investigates the mobility of the transcription factor Su(H) (referred to as CSL in the manuscript), mastermind (Mam), and Hairless under Notch-off and Notch-on conditions. The Notch-on state is enforced by ectopic expression of a constitutively active Notch protein (NΔECD). The study uses single molecule localization analysis to track tagged protein trajectories, and the analysis of trajectories relies on a tool called vbSPT (which uses variational Bayesian treatment of Hidden Markov Models). The authors observe that Notch-on conditions lead to a redistribution of CSL trajectories in which the proportion of CSL molecules showing restricted motion increases, and the number of faster diffusing CSL molecules decreases. The mobility distribution of CSL in the Notch-on state is (more-or-less) mirrored by Mam, whereas the mobility distribution of Hairless in the Notch-on state (which binds to Su(H) and exerts repressive effects) more closely resembles that of CSL in the Notch-off state. The manuscript is a bit dense and difficult to follow for the non-expert. Nevertheless, the work is informative and relevant because it provides a new line of direct evidence that the mobility of Su(H) and Mam becomes more restricted in the presence of active Notch, consistent with the (now intuitive) idea that the DNA-bound state for Su(H) is favored in the presence of nuclear NICD.

Major comments

- 1) For the naïve reader, it would be very helpful to demystify the vbSPT and DDMAP analyses more than has been done in the current manuscript. It would be helpful to clarify the rationale behind the choice of four "bins" of mobility in the vbSPT analysis (why not three, five, or a continuous distribution?). Perhaps showing the complete distribution of average diffusion coefficients for all trajectories might make it clear that there are natural divisions among mobility groups?
- 2) An alternative model might be one with two states - bound and free, with some trajectories capturing partial "occupancy" of each state (conversion from bound to free over the time of acquisition). This model could still be consistent with the example D2 and D3 trajectories shown in Fig. 1E, where the D2 trajectory has the molecule position fixed as in the D1 example for most of the frames but mobile in a couple of frames, and the D3 appears to have undergone two events that resulted in mobility restriction. In this alternative model, Notch-on simply shifts the distribution to one in which the bound state is more favorable. Is there evidence in the trajectory data to disfavor this alternative model?
- 3) (Related to Page 7, end of first paragraph) It is not possible based on the data to claim that Su(H) acquires the "Mam" mobility characteristics (even though this interpretation is likely and reasonable), because the mobility of Mam may also be influenced by the Notch state.
- 4) The magnitude of change in the anisotropy angle distribution plots is quite different in the vicinity of clusters, whereas in the Notch-off/Notch-on comparison of Figure 2 restricted to the D3/D4 group only, the difference in CSL angular distributions is much more muted. How is one to interpret these differences in magnitude? Is it that the E(spl) locus is somehow distinct, or that the bulk population includes a substantial fraction of molecules that are not associated with DNA, or ...?
- 5) In the conclusion section, the authors describe two likely molecular explanations that could contribute to "exploratory behaviour": DNA re-binding and protein "trapping". It might be helpful to represent these two molecular concepts in a bit more detail in the final model figure (5H).
- 6) The clustering analysis in Figure 5 is difficult to follow. It would be helpful to break the analysis down in a way that is more accessible to a non-expert in single molecule localization analysis.

Minor comment

- 1) It might be reasonable to refer to the transcription factor Su(H) in this manuscript because the system being studied is that of fly salivary gland cells.

2) The comparison between Notch-off and Notch-on is a steady state analysis between two different cultures of salivary gland cells - it might be helpful to note this more explicitly to aid the reader.

3) I agree with other reviewer comments about the uniqueness of the salivary gland model system - which the authors note in their limitations section.

Reviewer #2 (Comments to the Authors (Required)):

The manuscript from the Bray and Muresan groups, entitled "Changes in searching behaviour of CSL transcription complexes in Notch active conditions", builds upon their previous work and provides detailed in vivo single molecule localization microscopy data on the transcription factor CSL and its cofactors Mastermind, which is a transcriptional coactivator, and Hairless, which is a corepressor, under Notch on and off conditions. While the manuscript is largely descriptive, it does provide additional insights into the nuclear dynamics of CSL when Notch is active and inactive, and how these dynamics change when associated with Mastermind or Hairless. Importantly, the authors' data suggests an "exploratory behavior" exhibit by CSL under Notch on conditions and in regions of open chromatin. However, a limitation of the work is not exploring how presumably NICD confers this new behaviour to CSL, whereas the corepressor Hairless does not. Also, the authors rightly point out the limitations of their study, whereby they are using salivary glands from drosophila embryos, which contain polytene chromosomes. This experimental setup is advantageous for their microscopy studies, but may introduce artefacts that are not broadly applicable to all cell types. Nonetheless, the manuscript is well written and high-quality work and would be of interest to both researchers in the Notch field and those that study transcription factor (TF) dynamics. If the authors were to address my relatively minor criticisms summarize below, I would recommend their manuscript to be published in the journal Life Science Alliances.

1. On page 6, regarding the interpretation of Fig. 2A. I certainly agree that the largest difference between Notch on and off conditions is that the number of free CSL molecules (D4) decreases; however, the other conclusions seem a bit overstated and the changes look more modest, especially given the lack of statistical differences associated with those changes, e.g. D2 and D3. I would suggest rewriting that paragraph.

2. Page 8, "...Brownian trajectories near this region (Fig 3F, Fig S2C-D)". There isn't a Figure S2D.

3. In the conclusion, there is a brief mention that the TFs CTCF and CBX2 have been shown to display similar behavior as CSL, but I think it would be helpful to expand on these comparisons to really put into context how the nuclear dynamics of CSL are similar to some TFs but different from others, e.g. pioneer factors.

4. In Table S1, is it possible to include the Notch off data for Mastermind and Hairless?

Reviewer #3 (Comments to the Authors (Required)):

1. Baloul, Roussos et al use single molecule localisation microscopy to track the behaviors of individual molecules of different nuclear proteins of the Notch pathway transcriptional machineries: the transcription factor CSL/Su(H), the co-activator Mastermind, and the repressor Hairless.

Based on the distance and trajectory each molecule make during image acquisition, they describe different pools of CSL/Su(H) in the nuclei of Drosophila salivary gland cells: slow moving bound/trapped molecules, fast moving molecules (likely unbound), and molecules with an intermediate behaviour (main point 1). Importantly, these relative proportions change in the presence of the Notch pathway activator NΔExt favoring slow/intermediate behaviors at the expense of fast moving ones, suggesting that CSL/Su(H) is more restricted (likely bound to DNA) under Notch activation (main point 2).

They then focussed on a specific locus, known for its high Notch activity, the E(spl)-C. There, by zooming on the molecules located within 550nm of the E(spl)-C, they could show that around this locus, CSL/Su(H) adopts mainly a bound/slow moving behavior under Notch activation (main point 3).

Analyzing in more details the trajectories, they uncover that next to the E(spl)-C locus under Nocth activation, CSL/Su(H) molecules have a bias towards going backward, a behavior suggestive of a local searching mode where CSL/Su(H) would detach and re-attach to compatible sites on the DNA in the vicinity of the first site (main point 4).

Finally, looking more broadly, they could define potential CSL/Su(H) bound regions in the nucleus by looking at enrichment in slow moving molecules. They then show that in presence of Notch activation, CSL/Su(H) molecules around these clusters are more likely to have a backward movements (main point 5), suggesting some kind of confinement around bound CSL/Su(H) clusters.

These nice experiments bring thus experimental data to better model and understand how the CSL/Su(H) transcriptional machineries are built and how they find their target sites on the genome. The manuscript is clear and easy to read.

2. Points 1, 2, 4 and 5 are well supported by the data. Importantly, the difference between the data and the model proposed compatible with the data is clearly stated.

However, unless I missed something, point 3 needs more data to fully support the claim. Indeed, the E(spl)-C data presented is only about Notch-ON. It is thus impossible to evaluate whether Notch activation changes the behavior of CSL/Su(H) around the E(spl)-C: bound / not bound, Bayesian / Subdiffusive... Data collected in Notch-OFF conditions should be shown and analyzed, otherwise the text must be changed accordingly (for instance page 8 the sentences "As suggested by the trajectory maps, (Fig 3B) the ratio was significantly higher in Notch-On conditions, confirming that Notch activity promoted recruitment of CSL molecules to the target locus" or "The increase in binding of CSL at E(spl)-C in Notch-On condition" are not currently supported by the data presented in Figure 3.

Authors should also address whether Notch is active in the salivary gland they are probing. Indeed, it would be nice to see whether Notch is transcriptional active in the polyploid cells the authors probe. Authors should show that the Notch pathway, and thus the activity of the CSL/Su(H) transcription factor is modulated upon the addition of NΔExt; for instance imaging a Notch-pathway activity reporter, or qPCR for genes in the E(spl)-C locus. This piece of data would greatly strengthen all points of the manuscript linking more clearly the changes the authors observe in terms of CSL/Su(H) behaviours with the transcriptional activity of the complex.

3. Authors should discuss the results with Hairless whose behaviors in Figure 3E appear to follow the same pattern than those of CSL and Mam: more bound and less mobile next to E(spl)-C compared to regions away from E(spl)-C (only 4 nuclei analyzed based on points shown on the boxplots). How could this be interpreted in light of the repressive role of Hairless?

LSA-2023-02336-T Response to reviewers.**Reviewer #1 (Comments to the Authors (Required)):**

The work in this manuscript investigates the mobility of the transcription factor Su(H) (referred to as CSL in the manuscript), mastermind (Mam), and Hairless under Notch-off and Notch-on conditions. The Notch-on state is enforced by ectopic expression of a constitutively active Notch protein (NΔECD). The study uses single molecule localization analysis to track tagged protein trajectories, and the analysis of trajectories relies on a tool called vbSPT (which uses variational Bayesian treatment of Hidden Markov Models). The authors observe that Notch-on conditions lead to a redistribution of CSL trajectories in which the proportion of CSL molecules showing restricted motion increases, and the number of faster diffusing CSL molecules decreases. The mobility distribution of CSL in the Notch-on state is (more-or-less) mirrored by Mam, whereas the mobility distribution of Hairless in the Notch-on state (which binds to Su(H) and exerts repressive effects) more closely resembles that of CSL in the Notch-off state. The manuscript is a bit dense and difficult to follow for the non-expert. Nevertheless, the work is informative and relevant because it provides a new line of direct evidence that the mobility of Su(H) and Mam becomes more restricted in the presence of active Notch, consistent with the (now intuitive) idea that the DNA-bound state for Su(H) is favored in the presence of nuclear NICD.

We are glad the reviewer considers the work informative.

Major comments

1) For the naïve reader, it would be very helpful to demystify the vbSPT and DDMAP analyses more than has been done in the current manuscript. It would be helpful to clarify the rationale behind the choice of four "bins" of mobility in the vbSPT analysis (why not three, five, or a continuous distribution?). Perhaps showing the complete distribution of average diffusion coefficients for all trajectories might make it clear that there are natural divisions among mobility groups?

We thank the reviewer for highlighting that we had not explained the methods very clearly nor the rationale for the selection of 4 states for the subsequent analysis in vbSPT. We have now expanded the text introducing the methods (**page 4-5**) and explained that the 4 state model was selected as a better fit than those with fewer states, based on estimations from vbSPT models (**page 6**)

2) An alternative model might be one with two states - bound and free, with some trajectories capturing partial "occupancy" of each state (conversion from bound to free over the time of acquisition). This model could still be consistent with the example D2 and D3 trajectories shown in Fig. 1E, where the D2 trajectory has the molecule position fixed as in the D1 example for most of the frames but mobile in a couple of frames, and the D3 appears to have undergone two events that resulted in mobility restriction. In this alternative model, Notch-on simply shifts the distribution to one in which the bound state is more favorable. Is there evidence in the trajectory data to disfavor this alternative model?

The reviewer is correct that the behaviours of the molecules are composite and that broadly there is a continuum from those that are 100% bound to those that are 100% free. However, the two state model advocated by the reviewer is rejected when vbSPT performs model selection based on model likelihood. Indeed, it is very evident that the compact behaviours vary extensively and include some that are essentially stationary (usually considered to be specific binding) but others

where there is considerable movement in a small space (likely “trapped” within a chromatin domain and making less specific interactions). The use of the 4 state model, which is a better fit than 2 or 3 states, makes it possible to distinguish and compare some of the variety in these behaviours. We have added more explanation about this in the results on **page 5**.

3) (Related to Page 7, end of first paragraph) It is not possible based on the data to claim that Su(H) acquires the "Mam" mobility characteristics (even though this interpretation is likely and reasonable), because the mobility of Mam may also be influenced by the Notch state.

We apologise for the ambiguity and have rewritten this sentence on **page 7** “the Notch-induced changes in CSL behaviour are compatible with it being in a complex with Mam.....”

*4) The magnitude of change in the anisotropy angle distribution plots is quite different in the vicinity of clusters, whereas in the Notch-off/Notch-on comparison of Figure 2 restricted to the D3/D4 group only, the difference in CSL angular distributions is much more muted. How is one to interpret these differences in magnitude? Is it that the *E(spl)* locus is somehow distinct, or that the bulk population includes a substantial fraction of molecules that are not associated with DNA, or ...?*

The CSL angular distributions in the whole population will include a substantial fraction of molecules that are not associated with chromatin, as the reviewer suggests. In contrast, the analysis close to *E(spl)*-C focuses on the molecules that are associated with chromatin and compares their anisotropy to those elsewhere in the nucleus. We have modified the text in both sections of the results (**page 7** and **page 9**) to make this more explicit.

5) In the conclusion section, the authors describe two likely molecular explanations that could contribute to "exploratory behaviour": DNA re-binding and protein "trapping". It might be helpful to represent these two molecular concepts in a bit more detail in the final model figure (5H).

We have revised the model to represent these two concepts.

6) The clustering analysis in Figure 5 is difficult to follow. It would be helpful to break the analysis down in a way that is more accessible to a non-expert in single molecule localization analysis.

We thank the reviewer for highlighting this and have rewritten the paragraph in the results to explain the analysis in a more accessible way. We have also expanded the relevant section in the methods to give a more thorough and detailed explanation.

Minor comment

1) It might be reasonable to refer to the transcription factor Su(H) in this manuscript because the system being studied is that of fly salivary gland cells.

We use the term CSL (where the S stands for Su(H)) because people often tell us they find the name Suppressor of Hairless confusing (in part because there is a chromatin protein called Suppressor of Hairywing) and because it helps to link between species. We note that Judith Kimble also uses the term CSL in many of her papers on *C.elegans* Notch.

2) The comparison between Notch-off and Notch-on is a steady state analysis between two different cultures of salivary gland cells - it might be helpful to note this more explicitly to aid the reader.

We have added a sentence clarifying this point in the results on **page 6**.

3) I agree with other reviewer comments about the uniqueness of the salivary gland model system - which the authors note in their limitations section.

We have noted this limitation as the reviewer states. However, we also note that a number of fundamental core transcription mechanisms have been elucidated using this system and found to be universally important (e.g. see seminal papers from John Lis).

Reviewer #2 (Comments to the Authors (Required)):

The manuscript from the Bray and Muresan groups, entitled "Changes in searching behaviour of CSL transcription complexes in Notch active conditions", builds upon their previous work and provides detailed in vivo single molecule localization microscopy data on the transcription factor CSL and its cofactors Mastermind, which is a transcriptional coactivator, and Hairless, which is a corepressor, under Notch on and off conditions. While the manuscript is largely descriptive, it does provide additional insights into the nuclear dynamics of CSL when Notch is active and inactive, and how these dynamics change when associated with Mastermind or Hairless. Importantly, the authors' data suggests an "exploratory behavior" exhibit by CSL under Notch on conditions and in regions of open chromatin. However, a limitation of the work is not exploring how presumably NICD confers this new behaviour to CSL, whereas the corepressor Hairless does not. Also, the authors rightly point out the limitations of their study, whereby they are using salivary glands from drosophila embryos, which contain polytene chromosomes. This experimental setup is advantageous for their microscopy studies, but may introduce artefacts that are not broadly applicable to all cell types. Nonetheless, the manuscript is well written and high-quality work and would be of interest to both researchers in the Notch field and those that study transcription factor (TF) dynamics. If the authors were to address my relatively minor criticisms summarize below, I would recommend their manuscript to be published in the journal Life Science Alliances.

We are glad the reviewer considers the work high quality and provides new insights.

1. On page 6, regarding the interpretation of Fig. 2A. I certainly agree that the largest difference between Notch on and off conditions is that the number of free CSL molecules (D4) decreases; however, the other conclusions seem a bit overstated and the changes look more modest, especially given the lack of statistical differences associated with those changes, e.g. D2 and D3. I would suggest rewriting that paragraph.

We appreciate the comments of the reviewer and have now rewritten this part of the paragraph to avoid over-stating the results (**page 6-7**).

2. Page 8, "...Brownian trajectories near this region (Fig 3F, Fig S2C-D)". There isn't a Figure S2D. We thank the reviewer for pointing out this error which has been corrected.

3. In the conclusion, there is a brief mention that the TFs CTCF and CBX2 have been shown to display similar behavior as CSL, but I think it would be helpful to expand on these comparisons to really put into context how the nuclear dynamics of CSL are similar to some TFs but different from others, e.g. pioneer factors.

We thank the reviewer for this suggestion and have expanded the Discussion to include more reference to similarities and differences with other transcription factors and in particular how the results with CSL differ from those obtained for pioneer factors (**page 12**)

4. In Table S1, is it possible to include the Notch off data for Mastermind and Hairless?

These have been added.

Reviewer #3 (Comments to the Authors (Required)):

1. Baloul, Roussos et al use single molecule localisation microscopy to track the behaviors of individual molecules of different nuclear proteins of the Notch pathway transcriptional machineries: the transcription factor CSL/Su(H), the co-activator Mastermind, and the repressor Hairless. Based on the distance and trajectory each molecule make during image acquisition, they describe different pools of CSL/Su(H) in the nuclei of *Drosophila* salivary gland cells: slow moving bound/trapped molecules, fast moving molecules (likely unbound), and molecules with an intermediate behaviour (main point 1). Importantly, these relative proportions change in the presence of the Notch pathway activator $N\Delta Ext$ favoring slow/intermediate behaviors at the expense of fast moving ones, suggesting that CSL/Su(H) is more restricted (likely bound to DNA) under Notch activation (main point 2). They then focussed on a specific locus, known for its high Notch activity, the *E(spl)-C*. There, by zooming on the molecules located within 550nm of the *E(spl)-C*, they could show that around this locus, CSL/Su(H) adopts mainly a bound/slow moving behavior under Notch activation (main point 3). Analyzing in more details the trajectories, they uncover that next to the *E(spl)-C* locus under Nocth activation, CSL/Su(H) molecules have a bias towards going backward, a behavior suggestive of a local searching mode where CSL/Su(H) would detach and re-attach to compatible sites on the DNA in the vicinity of the first site (main point 4). Finally, looking more broadly, they could define potential CSL/Su(H) bound regions in the nucleus by looking at enrichment in slow moving molecules. They then show that in presence of Notch activation, CSL/Su(H) molecules around these clusters are more likely to have a backward movements (main point 5), suggesting some kind of confinement around bound CSL/Su(H) clusters. These nice experiments bring thus experimental data to better model and understand how the CSL/Su(H) transcriptional machineries are built and how they find their target sites on the genome. The manuscript is clear and easy to read.

We thank the reviewer for their positive comments about our manuscript.

2. Points 1, 2, 4 and 5 are well supported by the data. Importantly, the difference between the data and the model proposed compatible with the data is clearly stated. However, unless I missed something, point 3 needs more data to fully support the claim. Indeed, the *E(spl)-C* data presented is only about Notch-ON. It is thus impossible to evaluate whether Notch activation changes the behavior of CSL/Su(H) around the *E(spl)-C*: bound / not bound, Bayesian / Subdiffusive... Data collected in Notch-OFF conditions should be shown and analyzed, otherwise the text must be changed accordingly (for instance page 8 the sentences "As suggested by the trajectory maps, (Fig 3B) the ratio was significantly higher in Notch-On conditions, confirming that Notch activity promoted recruitment of CSL molecules to the target locus" or "The increase in binding of CSL at *E(spl)-C* in Notch-On condition" are not currently supported by the data presented in Figure 3.

We thank the reviewer for highlighting this omission. We have in fact performed all of the imaging and analysis in Notch-Off conditions but were concerned it could be confusing to include all those data also. We have now added all of the results for Notch-Off namely:

- Notch-Off and Notch-On density maps from representative nuclei for CSL, Mam Hairless in **New Figure 3A**
- Table summarizing enrichments at *E(spl)*-C in **new Figure 3C**
- Summary of Mam and Hairless characteristics in Notch-Off (**in new supplementary Tables S1A,B and S2A,B**)

These data clearly show that CSL and Mam increase at *E(spl)*-C in Notch-On conditions.

*Authors should also address whether Notch is active in the salivary gland they are probing. Indeed, it would be nice to see whether Notch is transcriptional active in the polyploid cells the authors probe. Authors should show that the Notch pathway, and thus the activity of the CSL/Su(H) transcription factor is modulated upon the addition of NΔExt; for instance imaging a Notch-pathway activity reporter, or qPCR for genes in the *E(spl)*-C locus. This piece of data would greatly strengthen all points of the manuscript linking more clearly the changes the authors observe in terms of CSL/Su(H) behaviours with the transcriptional activity of the complex.*

In previous and related publications we have performed qPCR and smFISH to measure Notch target gene expression under the conditions described here and shown that there is no/very low expression in Notch-Off and that expression is significantly induced in Notch-On. We now state this explicitly in the text **on page 6** and refer to those publications (Gomez-Lamarca et al, 2018; deHaro-Arbona et al, 2023).

*3. Authors should discuss the results with Hairless whose behaviors in Figure 3E appear to follow the same pattern than those of CSL and Mam: more bound and less mobile next to *E(spl)*-C compared to regions away from *E(spl)*-C (only 4 nuclei analyzed based on points shown on the boxplots). How could this be interpreted in light of the repressive role of Hairless?*

We agree with the reviewer that the observations relating to Hairless are interesting. As discussed in our previous publication, a likely explanation for the increase in Hairless recruitment is that the chromatin in the gene region becomes more accessible/ reduced nucleosome density enabling binding also of CSL complexed with Hairless. Functionally its recruitment could enable a more rapid shutdown, when levels of active Notch decline and/or it could function as an amplitude rheostat. The phenotypes associated with Hairless loss of function, (limited and local derepression of target genes) are very compatible with these models. A paragraph summarising these points has been added to the Discussion on **page 12**.

November 27, 2023

RE: Life Science Alliance Manuscript #LSA-2023-02336-TR

Prof. Sarah J. Bray
University of Cambridge
Dept. of Physiology, Development and Neuroscience
Downing Street
Cambridge, CB2 3DY
United Kingdom

Dear Dr. Bray,

Thank you for submitting your revised manuscript entitled "Changes in searching behaviour of CSL transcription complexes in Notch active conditions". We would be happy to publish your paper in Life Science Alliance pending final revisions necessary to meet our formatting guidelines.

-please add ORCID ID to the secondary corresponding author -- they should have received instructions on how to do so

A. FINAL FILES:

B. MANUSCRIPT ORGANIZATION AND FORMATTING:

****It is Life Science Alliance policy that if requested, original data images must be made available to the editors. Failure to provide original images upon request will result in unavoidable delays in publication. Please ensure that you have access to all original**

data images prior to final submission.**

The license to publish form must be signed before your manuscript can be sent to production. A link to the electronic license to publish form will be available to the corresponding author only. Please take a moment to check your funder requirements.

Sincerely,

Reviewer #1 (Comments to the Authors (Required)):

The revised manuscript has done a superb job responding to reviewer comments, and the authors should be congratulated for an excellent body of work, so nicely presented.

Reviewer #2 (Comments to the Authors (Required)):

The authors have satisfactorily addressed the criticisms I raised with their original submission. I recommend that the revised version of the manuscript is published in LSA and I congratulate the authors on a very nice piece of work.

Reviewer #3 (Comments to the Authors (Required)):

The authors replied satisfactorily to all my comments and amended the text where I thought necessary. I have no further comments and support publication of this nice study.

December 4, 2023

RE: Life Science Alliance Manuscript #LSA-2023-02336-TRR

Prof. Sarah J. Bray
University of Cambridge
Dept. of Physiology, Development and Neuroscience
University of Cambridge
Downing Street
Cambridge, cambs CB2 3DY
United Kingdom

Dear Dr. Bray,

Thank you for submitting your Research Article entitled "Changes in searching behaviour of CSL transcription complexes in Notch active conditions". It is a pleasure to let you know that your manuscript is now accepted for publication in Life Science Alliance. Congratulations on this interesting work.

DISTRIBUTION OF MATERIALS:

Again, congratulations on a very nice paper. I hope you found the review process to be constructive and are pleased with how the manuscript was handled editorially. We look forward to future exciting submissions from your lab.

Sincerely,
